# The Impact of Motherhood on Women’s Career Progression: A Scoping Review of Evidence-Based Interventions

**DOI:** 10.3390/bs14040275

**Published:** 2024-03-26

**Authors:** Ana Júlia Calegari Torres, Letícia Barbosa-Silva, Ligia Carolina Oliveira-Silva, Olívia Pillar Perez Miziara, Ully Carolina Rodrigues Guahy, Alexandra N. Fisher, Michelle K. Ryan

**Affiliations:** 1Institute of Psychology, Federal University of Uberlândia, Av. Pará, 1720-Bloco 2C, Av. Maranhão, s/n-Campus Umuarama, Uberlândia 38405-240, MG, Brazil; leticiabarbosasilva4@ufu.br (L.B.-S.); ligiacarolina.oliveira-silva@anu.edu.au (L.C.O.-S.); olivia.miziara@ufu.br (O.P.P.M.); 2The Global Institute for Women’s Leadership, Australian National University, Canberra, ACT 2600, Australia; alexandra.fisher@anu.edu.au (A.N.F.); michelle.ryan@anu.edu.au (M.K.R.); 3Institute of Psychology, University of São Paulo, Av. Professor Mello Moraes, 1721-Butantã, São Paulo 05508-030, SP, Brazil; uguahy@gmail.com; 4The Faculty of Economics and Business, University of Groningen, Nettelbosje 2, 9747 AE Groningen, The Netherlands

**Keywords:** motherhood, interventions, review, career, progression, leadership

## Abstract

(1) Background: Despite the progress made by women in the workplace, mothers still face systemic barriers that prevent them from advancing professionally. This “motherhood penalty” involves a variety of discriminatory practices and experiences that mothers can face at work, including being held to stricter standards regarding salary and recruitment. Despite ongoing research on the association between motherhood and career outcomes, few studies specifically explore how motherhood impacts career advancement and, consequently, access to leadership. This scoping review seeks to gain an understanding of how motherhood impacts women’s career progression, and how interventions can address the underrepresentation of mothers in leadership. (2) Methods: Following the PRISMA-ScR framework, we analyzed 52 articles from 2010 to 2022, drawn from 10 databases. (3) Results: The results showed both negative and positive impacts of motherhood on career progression, affecting mothers’ attitudes, feelings, and behaviors and yielding changes in interpersonal relationships and work conditions. Intersectionality is highlighted, urging a nuanced examination of challenges faced by mothers from a diversity of backgrounds. Recommendations for interventions include individual and institutional efforts, comprising societal support structures, organizational policy changes, and cultural shifts. (4) Conclusions: This scoping review offers an updated perspective on a classic challenge, providing practical insights for a more inclusive and structural understanding of the career trajectories of working mothers.

## 1. Introduction

In the workplace, mothers face a motherhood penalty, where they are perceived as being unfit for leadership roles, are evaluated as less competent and less committed to their careers, receive lower salaries, and are denied advancement opportunities [1,2,3]. The experience of motherhood often comes with a lower sense of control and belonging, followed by elevated productivity demands [4]. As a consequence, mothers tend to occupy fewer leadership positions than men (including fathers) and childfree women.

This penalty is, at least in part, due to gendered norms and cultures inherent in career development policy and practice, which still focuses on men’s experiences, values male attributes, and reinforces traditional notions of leadership [4]. Yet, to date, there has not been, to our knowledge, a systematic examination of how motherhood impacts career progression and access to leadership. Additionally, just describing the problem is not enough, as we also need to visualize which and how interventions can offer support to mothers through multidimensional sources. This should include not only their behavior as employees, but also how organizational culture and government policies, for instance, can boost mothers’ career progression and hopefully pave their way towards leadership positions.

In the current research, we aim to fill this gap by conducting a scoping review of the literature to understand (1) how motherhood impacts women’s career progression and (2) how interventions can address the underrepresentation of mothers in leadership roles. In doing so, this review will offer valuable insight into how to reimagine leadership with motherhood in mind.

## 2. Barriers to Leadership for Mothers

Although notions of “mothering” are widely acclaimed and valued throughout society, difficulties in finding balance between career and motherhood remain, resulting in career breaks and reduced work hours [4,5]. Cross-country research demonstrates that there is a decline in mothers’ participation in the workforce after childbirth. On average, 24% of women exit the labor market in their first year of motherhood. Five years later, the percentage drops to 17% and after a decade, still 15% are absent [6]. Considering the rather unquestioned expectation for continued work throughout the years to favor career progression, not surprisingly, mothers remain markedly underrepresented in leadership positions.

Research has offered several explanations for why this might be. The first is stereotypes about gender and leadership [7]. Leadership remains largely associated with stereotypically masculine attributes such as competitiveness, aggression, and selfishness [8], contributing to a scenario where women are viewed as less equipped to assume leadership roles [9]. This phenomenon, known as ‘role incongruity’ [10], highlights the expected inconsistencies between the traditional female gender role and the leadership role. The equation of leadership with masculinity means that women and gender-diverse people who do not conform to these stereotypic expectations are at a disadvantage when applying for leadership roles, as they are less likely to be seen as typical ‘leaders’. This is especially true for mothers, as the qualities expected of a mother, such as warmth and affection, are directly at odds with the stereotypical qualities expected of a leader [10,11].

The demands of motherhood may also affect the ability of mothers to perform in leadership roles. The motherhood penalty comes with an assumption that mothers are less committed or competent, which implies promotion delays, limited career options, or the need to make potentially career-harming decisions to meet children’s needs [12]. With the current economic model of work, the rooted idea of the ‘ideal employee’ implies a person without domestic and family responsibilities who can devote long hours to uninterrupted work [13,14]. This assumption often presumes that another person, usually a woman, is at home taking care of household chores and caring responsibilities, whereas men are expected to perform the majority of financial and provider responsibilities within their households [15,16]. Gender roles within heterosexual relationships reinforce this model, such that mothers often end up working a taxing ‘double shift’ to manage the demands of both home and career, which may even lead to lower productivity and poor mental health [17,18].

Women with children are, therefore, caught between a rock and a hard place where to be a good mother, they must prioritize their families over their careers, but to be a good leader, they must prioritize their careers over their families. Unable to accommodate such contradictory expectations, mothers commonly deviate from the traditional career path, taking career breaks, reducing working hours, or even leaving the workforce indefinitely, all of which negatively affects their chances of career advancement. This may cause mothers to forgo leadership opportunities or even cut back at work in an effort to be a ‘good’ mother. When they decide to return to the paid workforce, mothers are less likely to be interviewed, hired, or promoted, and they receive lower salaries [18]. The incompatibility between career and family life can lead to a myriad of negative consequences for working mothers, including feelings of guilt, emotional overload, and fatigue, as the requirement to perform both roles properly is incompatible with reality [19,20].

Taken together, these situations create a maternal wall, composed of formal and informal forms of discrimination that hinder women’s professional advancement after having children [21]. Understanding how motherhood influences career advancement and the path to leadership attainment is therefore crucial to creating inclusive and supportive environments for paid working mothers. Additionally, acknowledging these impacts can better inform organizations regarding how to develop interventions which recognize the multidimensional (individual, interpersonal, organizational, and societal) nature of the issue.

## 3. The Need for Interventions to Address the Underrepresentation of Mothers in Leadership

Interventions encompass a broad spectrum of activities, treatments, programs, or initiatives designed to tackle specific issues, achieve defined outcomes, or influence particular variables [22]. Considering the specific barriers that motherhood imposes on women’s career advancement, gender- and motherhood-sensitive interventions may offer important opportunities for leveling the playing field if they aim to reduce gender inequality and transform social conditions by addressing structural issues such as biases and stereotypes [7].

Interventions aimed at transforming traditional notions of leadership are of particular importance. Therefore, rethinking leadership in the context of motherhood can be a strategic imperative for fostering environments that thrive on diverse approaches to problem-solving. These targeted efforts can help reconcile paid work and other roles assumed by women, promoting a more supportive environment for working mothers.

## 4. Current Study

Considering that mothers face systematic barriers to leadership positions, it is important to analyze how motherhood may influence women’s professional ascension and the kinds of interventions that may help their path towards leadership. To initially map the issue, we first conducted a rapid review of the literature about motherhood and leadership, which revealed a scarcity of research about mothers in formal leadership positions and a particular lack of research that considers the role of interventions. Due to this paucity, the search strategies were expanded to encompass a broader scope through the use of the term “career progression” instead of “leadership”. A new rapid search was accomplished, which evidenced the feasibility of a scoping review to analyze career progression instead of leadership.

Career progression extends beyond mere leadership and encompasses a broader spectrum of career aspects, including various professional advancements that may not necessarily involve formal leadership roles. Indeed, both the prevailing definitions of career advancement and leadership often perpetuate a male-centric viewpoint, concentrating solely on high-ranking executive positions rarely held by mothers. Adopting a broader perspective allowed us to recognize and embrace the diversity inherent in professional success, considering the varied career trajectories that women, particularly mothers, may pursue.

Although the literature has advanced in terms of analyzing how motherhood may impact career progression, to our knowledge, there is no systematic literature review on this topic, or even on interventions in this context. This gap is concerning, as the design of actions and strategies to foster mothers’ economic and labor participation need to be grounded on a solid foundation of evidence-based research. To achieve this, there is a need to explore and map the state of the art on the topic, which can be done through a scoping review. Additionally, a scoping review allows for identifying the current gaps in the literature, which is paramount for guiding research agendas. By mapping how the latest research has investigated the impact of motherhood and career progression, we expect to offer systematized evidence for fostering future research and interventions aimed to help mothers ascend professionally. The following research questions were investigated:

(1) How does motherhood impact women’s career progression?

(2) How can interventions address the underrepresentation of mothers in leadership?

## 5. Method

### 5.1. Study Design and Report Guidelines

We conducted this scoping review based on Arksey and O’Malley’s [23] five-step methodological framework: (1) identifying pertinent research questions, (2) conducting a comprehensive search for studies relevant to our research questions, (3) employing a systematic study selection process with predetermined eligibility criteria, (4) charting relevant data from included studies using Rayyan as a data extraction tool, and (5) summarizing and reporting the results. We also employed the preferred reporting items for systematic reviews and meta-analyses extension for scoping reviews [24], and four of the authors (AJ, LB, OM, UG) assessed the papers for quality according to PRISMA-ScR checklists.

### 5.2. Search Strategy and Databases

Our search strategy was grounded in the peer review of electronic search strategies checklist [25]. We used the following databases: Lilacs, Pepsic, Scielo, PsycInfo, EBSCO, Web of Science, Scopus, MEDLINE/Pubmed, EMBASE, and CENTRAL. We chose ten databases, in line with best practice [26], and due to their coverage of most publications on the topic. We also included three Brazilian databases (Lilacs, Pepsic, Scielo), considering the nationality of most of the authors, the fact that they are open access, and that numerous Brazilian authors publish in English in the Brazilian journals that are featured in these respective databases. We filtered peer-reviewed articles from 2010 to 2022, including publications in English. We used the search string for the title, abstract, and keywords fields.

We used the population (working mothers), concepts (motherhood, career progression), and context (workplaces, academy, and organizations) framework to define the search terms. After the refinements, we used the following string for the search: (“parental role” OR “mother” OR “mothers” OR “motherhood” OR “mom” OR “moms”) AND (“career pathways” OR “career trajectory” OR “career breaks” OR “career interruptions” OR “child penalties” OR “motherhood penalty” OR “motherhood bias” OR “career success” OR “career achievement” OR “career progression” OR “career advancement” OR “professional advancement” OR “job progression” OR “job advancement” OR “work advancement” OR “work progression” OR “maternal wall”).

### 5.3. Eligibility Criteria and Study Selection Process

Our eligibility criteria were: (a) articles published in peer-reviewed journals; (b) articles written in English; (c) full-text availability; (d) publication year between 2010 and 2022; (e) quantitative, qualitative, or mixed methods; (f) samples should include women; (g) title, objective, and/or variables should address motherhood, career advancement, and interventions. We analyzed only articles in English due to limitations of time and resources, and the choice to focus on a 10-year timeframe was made in order to provide a contemporary overview of the topic. Articles with mixed-gender samples were included as long as they had separate data on mothers.

We uploaded the articles retrieved from the databases to Rayyan [27], with duplicates excluded and the initial screening of titles and abstracts conducted by three of the authors, coded as AJ, LB, and UG. After the screening process, we fully read the remaining articles, which were independently evaluated by the same three authors. A fourth researcher helped the authors reach a consensus.

We first conceptualized our research questions and purpose based on a rapid review of the motherhood and leadership literature. The rapid review revealed that there is little research about mothers in leadership positions, so using career progression as a term proved to be more comprehensive. Afterwards, we elected a population, concepts, and context (PCC) framework to improve the research question and purpose of this study:

Population: working mothers over 18 years old;

Concept: motherhood, career progression, leadership;

Context: academia and organizations.

### 5.4. Data Extraction and Synthesis

Four of the authors independently conducted data extraction using a standardized data extraction sheet. The extracted information encompassed author details, publication dates, journal main nationality, first authors’ nationality, aims, study type (qualitative, quantitative, or mixed methods), research design (experiment, case study, longitudinal, or cross-sectional), data collection method, participant sample, context, intervention type (if applicable), main results, conclusions, and key findings that relate to the research questions.

### 5.5. Study Selection Process

The search and selection of articles is detailed in Figure 1. Initially, we found 895 articles in the 10 selected databases, but after excluding duplicates, we analyzed 413 articles based on their titles, abstracts, and keywords. In the eligibility analysis, we fully read 91 articles, and the final sample consisted of 52 articles that met all the inclusion criteria. The inter-rater agreement rate was 49.5% between the three authors that worked as judges (AJ, LB, UG).

## 6. Results

### General Characteristics

Of the 52 articles we included, 48.1% were quantitative, 34.6% were qualitative, and 17.3% used mixed methods. Most used a cross-sectional design (53.8%) and surveys as the data collection method (63.5%). The highest percentage of author nationality, regarding the university of affiliation, was the USA (28.8%), while most of the journals were based in the United Kingdom (44.2%), according to information extracted from the Scimago Journal & Country Rank. The journal with the most publications (9.6%) was Gender, Work & Organization. Regarding the year of publication, 2021 was the year with the most papers (11), followed by 2022 (7). Finally, most of the studies, 55.8%, used a female-only sample, the research was conducted in organizations (51.9%), and it generally did not distinguish between the knowledge field of the participants (61.5%; see Table 1 for details).

We gathered and analyzed further categories addressing the main findings of each paper to answer the two research questions. More detail on the findings for each question are available in Appendix A.

## 7. Discussion

### 7.1. Question 1: How Does Motherhood Impact Women’s Career Progression?

#### 7.1.1. More Barriers to Leadership for Mothers

Unsurprisingly, the reviewed literature reinforced that women are still less likely to occupy a leadership position after becoming mothers and indicated multiple barriers to women’s leadership and career progression more generally [28,29]. Our analysis evidenced that many of the barriers that women face with regard to their career progression are treated as internal, personal choices rather than structural problems, e.g., references [30,31,32].

The most frequent internalized barrier had to do with the societal expectations that mothers must put the needs of others before their own, sacrificing their own interests and career aspirations to fulfill motherly duties, e.g., references [33,34,35]. Mothers still face identity conflicts caused by unrealistic ideals of what is a good mother and a successful worker [36,37], in addition to the emotional challenge of being mostly responsible for what their children will become [38]. Women declined promotion offers, in part, because of the double burden caused by the accumulation of family and leadership responsibilities, e.g., references [30,32,33].

A lower sense of control over professional life was identified, followed by negative feelings like stress and guilt, lower satisfaction with their achievements, more fear of job loss, and a lower sense of entitlement to use benefits such as reduced hours or maternity leave [35,39,40,41,42]. Additionally, mothers experienced a lower sense of belonging at work, as they often had to prove themselves by putting more effort in to achieve higher positions than their childfree counterparts [30].

In contrast to the focus on internalized barriers such as self-blaming and self-management of motherhood, other studies analyzed more contextual aspects. Time availability was identified to be one of the major contextual barriers to mothers in senior positions or leadership aspirants, as caregiving responsibilities put additional constraints on their time and availability [31,35]. Compared to fathers, mothers are more limited in terms of their availability to participate in events outside of work hours or to travel for business [43]. Therefore, the evidence shows they are disregarded from positions that require extended time away from home, a common demand for many leadership roles [44,45]. As a consequence, some studies emphasized the monetary losses post-motherhood, focusing on wage penalties, incoming losses due to reduced working hours, and transitions to lower-earning occupations [31,46].

Motherhood was also associated with lower job retention due to barriers like poor access to childcare or other work–life balance challenges, e.g., references [31,32]. In some cases, mothers reported experiencing career interruptions in the form of returning later than expected to work, resorting to informal, flexible, or part-time work, postponing promotion opportunities, or even dropping out of the workforce indefinitely, e.g., references [28,33]. Other papers addressed how exclusively female parental leave policies contribute to reducing women’s career opportunities, professional networks, and partnerships, e.g., references [30,36,47].

#### 7.1.2. The Weight and Endurance of the Motherhood Penalty

The analyzed articles demonstrated the persistence of the motherhood penalty, as they documented the ways in which mothers are impacted by notions of the ‘ideal worker’ who is free from domestic responsibilities and able to work overtime whenever needed. Such stereotypes remain the default assumption in many workplaces and are in direct conflict with societal expectations of mothers, implying they should be the primary caregivers and homemakers within their families [11]. The consequences are that mothers experience elevated turnover rates, frequent employment transitions, low-paying positions, and a lower likelihood of being recommended for hiring in comparison to men [47,48,49,50,51].

The motherhood penalty also evidences how mothers are commonly devalued and subject to scrutiny and doubts over their commitment to their work. The belief that women “use” pregnancy or motherhood to avoid work still exists [38]; mothers face constant discrimination and have their personal life as a current topic of debate at workplaces, in addition to being labeled as “the pregnant one” or “the one with the baby” [52]. Women may be penalized just by achieving childbearing age, corroborating the negative impact that even the prospect of motherhood may have on their careers.

Due to the often unrealistic expectations of time availability and flexibility in leadership positions, the motherhood penalty contributes to the perception that mothers are unsuitable for leadership roles. Benard and Correll’s [48] study, for instance, showed how mothers or pregnant women are rated as less competent, less committed, held to stricter standards, and penalized in salary and hiring decisions when compared to their childless counterparts. Mothers are also assigned to less interesting tasks, are not seen as suitable for management positions, and have to struggle to maintain their credibility [18,53]. Similar results were found by Thébaud and Taylor [51] and Schlehofer [52], which show how mothers are subjected to reduced competency perceptions and are viewed as less hirable.

As if it was not enough, there was also evidence of a dominance penalty for those who excel during motherhood: highly successful mothers were rated as less likable and warm, which can lead to penalization in other forms such as salary, hiring, and other organizational rewards, e.g., references [54]. Therefore, we can say that research in the last 10 years attests to how society pressures mothers to bear the brunt of responsibilities around childcare and family life, often compromising their career aspirations and progression.

#### 7.1.3. The Motherhood Advantage?

In contrast to the penalties outlined above, some of the reviewed articles highlighted positive career outcomes linked to motherhood, as the experience of motherhood is many-sided and has the potential to promote new skills development. For instance, women reported improved work relationships, perceiving greater appreciation by their colleagues as working mothers than they did before having children [51], and more efficient time management and problem-solving techniques [55]. Other studies indicated women felt greater motivation to complete different work responsibilities after becoming mothers, such as finishing coursework, as well as an increase in cognitive knowledge, seen to be a result of coping with motherhood challenges such as taking care of children and multitasking to complete household chores [32,43,55,56].

One article in particular showed a wide range of positive impacts of motherhood for job performance and career advancement in the U.S. tourism and hospitality industry [55]. This study highlighted how working mothers may experience more courage and increased confidence to pursue positions involving management and leadership roles. Mothers were reported to have improved willpower and emotional intelligence, which included more patience, tolerance, positive emotions and attitudes, confidence, and a stronger mindset.

As for more tangible outcomes, unexpectedly, Ma [56] and Magnusson [43] showed that wages can be positively impacted by parenthood for men and women, as they found that after childbirth, average wages were higher for married/cohabiting respondents with children for fathers and mothers. Regarding career progression, Morgenroth et al. [18] found that stereotypes of mothers are seen as more similar to stereotypes of managers and ideal managers than stereotypes of women in general, evidencing a benefit for mothers in terms of stereotype content. Therefore, despite working mothers experiencing more negative than positive impacts overall, the reviewed studies demonstrate how motherhood can also bring certain benefits to career progression.

### 7.2. Question 2: How Can Interventions Address the Underrepresentation of Mothers in Leadership?

#### 7.2.1. Can Individual-Focused Strategies Work as Effective Interventions?

Some of the interventions we identified in the career literature focus on individual-focused strategies; that is, women’s individual efforts and agency to manage their careers. Career counseling is one example, which was suggested to help mothers in assessing career goals, developing strategies, and making informed career decisions [57]. Likewise, therapeutic support services such as individual counseling or support groups are also indicated to assist them to address internalized stigma, navigate societal expectations, and develop coping strategies to deal with the challenges of balancing motherhood and career.

Skills development was identified as another personal strategy to reduce family–work conflict and maintain professional relevance even if mothers choose to slow down their career progression temporarily. Among these, time management skills were proposed to facilitate the balance between motherhood and career demands [58]. Learning negotiation skills, in turn, was proposed to support mothers in discussing working conditions with their supervisors or human resources [59].

Seeking further education was a commonly suggested strategy for tackling career progression barriers. Education on women’s rights and labor laws, for instance, were recommended to make women aware of their rights and legal protections against discrimination at work [46,60]. Financial literacy education was also proposed as something important for mothers, since budgeting, savings, and investment strategies could help them navigate career transitions, especially during economic crises [32,44].

What these strategies have in common is that they all focus on individual efforts women can pursue—which usually involve changing themselves or their behavior—to “compensate” workplaces for their motherhood. Such individual strategies may be useful to a degree, but they do not acknowledge the societal, cultural, and economic barriers women face. In their review, Ryan and Morgenroth [61] classify such interventions as those that try to provide women with the ‘right’ tools and skills to achieve leadership positions. However, these may end up being problematic due to a number of reasons such as the reinforcement of gender stereotypes, which indirectly blame women for inequalities and fail to address the root of gender inequalities. Women, especially mothers, should not be solely held accountable for changing a system that inherently perpetuates inequality and discrimination.

Beyond the interventions identified in the reviewed articles, recommendations from Oliveira-Silva and Barbosa-Silva [7] emphasize that interventions for gender equality should be multilevel and target systemic sexist structures, such as the gender norms and stereotypes that contribute to perpetuating women’s main and naturalized role in reproduction, nurturing, and child rearing. When transposing these recommendations to motherhood and career, they would include actions at the interpersonal level, such as discussing the desired number and spacing of children with a partner, how to share and balance family priorities, and the impacts of postponing pregnancies in favor of a career—a dilemma so familiar for women but so unknown to men.

Although not clearly identified in the reviewed literature, we also highlight the importance of adopting an intersectional perspective, such that identity markers (e.g., race, ethnicity, sexual orientation, or socioeconomic status) and the specific conditions of mothers (e.g., number and age of the children, marital status, and support network availability) are considered. Therefore, we should be aware of interventions that imply we need to “fix” women [61]; instead, interventions should aim to address the broader societal structures by changing workplace and government policies, culture, and practices.

#### 7.2.2. Interventions to Restructure How We See Motherhood and Leadership

Interventions should challenge traditional gender norms and lead to a cultural shift that redefines societal perceptions of motherhood and leadership. It is important to celebrate the diverse roles women can occupy, recognizing they can be whatever they want—including leaders. To make this possible, we need to tackle stereotypical and unreasonable views of what constitutes a “good” mother and a “successful” leader.

Valuing care work and promoting an equitable division of family duties is essential, as it contributes to contesting the notion that women should be the primary caregivers. This also implies recognizing the socioeconomic significance, effort, and skills involved in taking care of children and elderly people [62]. Although the following interventions do not exhaust the needs and barriers faced by mothers in their careers, the combined efforts of individuals, organizations, and governments are necessary to tackle some of the unequal structures that still hinder women’s decisions professionally and personally.

##### It Takes a Village to Raise a Child

Overall, cultivating connections and counting on the support of other people is essential for creating an environment where mothers feel acknowledged and empowered while also respecting their multifaceted identities. Several articles in our sample, e.g., references [44,58], indicated the support of family and friends as a key enabler for mother’s career progression.

Therefore, career arrangements that allow for the equal sharing of family responsibilities with spouses is a must, and it is important for couples to have open communication, outlining responsibilities and expectations with a clear understanding of each partner’s responsibilities, as highlighted by part of the reviewed articles, e.g., references [36,37]. Additionally, we propose that having immediate family members (such as parents, siblings, or friends) close by and adding to the support network also represent meaningful boosters to women’s career advancement. Such community and family support is especially critical for the career progression of single mothers.

##### The Importance of Women Helping Other Women

Based on the findings of Bowyer et al. [30] and Eren [36], we propose that interventions involving role models may be especially helpful, as getting to know other stories could support women in navigating the duality of work and motherhood when exploring different career paths. Networking is also important, and it can be done by actively engaging with other parents (within or outside the workplace) to build a peer support network to share experiences and career and family related advice. Within the workplace, this network may be an important source of feedback for mothers regarding their skills and contributions, since they may show a tendency toward understatement and poor self-assessment, as shown by Schueller-Weidekamm and Kautzky-Willer’s results [58].

Schueller-Weidekamm and Kautzky-Willer [58] and Kristensen et al. [63] propose that organizations can also invest in mentorship programs, succession planning, or leadership programs that promote female talent and emphasize collaborative leadership styles, challenging the traditional views of what makes a good leader. For best practices, in addition to providing instrumental and psychosocial support for mothers through mentoring, workplaces can stimulate formal women’s platforms and networks to foster connection and sharing experiences [44,55].

##### Focusing on Workplace Arrangements Makes a Difference

Regarding organizational interventions, they need to go beyond the usual “leave” policies and also provide “return” and “stay” policies; an approach that is often underestimated. In addition to extended parental leaves (which are indeed important), organizations should also provide a safe environment to which women would like to return, rather than a work structure that maintains the status quo and penalizes them for trying to balance family responsibilities and a career. One way to achieve equality for mothers is to create opportunities for them to continue working while also raising their children, enabling a real integration between work and family, which requires profound changes in workplace practices [48,64].

Therefore, organizations should focus on flexible policies and work arrangements in response to the changing family circumstances of their employees, such as adjusting working hours, offering remote work options, or avoiding planned meetings late at night. They could also enable job redesigns and temporarily restructure the roles of pregnant women or new mothers by assigning them administrative work and less travel, as suggested by Whittington [28].

Based on the common negative impacts identified in the reviewed articles, we propose that inclusive events and activities, child-friendly spaces, and breastfeeding rooms are other good initiatives to promote a more children-friendly work environment. Maternity leave is one of the main reasons for women’s career breaks and may be a “burden” to mothers according to Maxwell et al. [59], while fathers often continue to work and advance professionally regardless of their family situation. Therefore, parental leave should be equally shared, allowing families enough time to bond with newborns and achieve a more equitable distribution of caregiving responsibilities. Organizations can facilitate this by encouraging and supporting men to participate equally in caring roles by reframing caregiving in ways that facilitate agentic goals and normalize men’s participation in these roles [65]. Male managers and leaders who role model equal parenting may also help to achieve these aims. Organizational and governmental policy that introduces a well-paid, nontransferable “use it or lose it” policy for parental leave of fathers and secondary caregivers may also help to normalize men’s equal participation in caregiving [66], as well as initiatives that address men’s own gender role attitudes and biases [67].

Flexible work arrangements should also be combined with family-friendly benefits, like extended parental leave, on-site childcare facilities, and the possibility of taking days off for caregiving reasons. Considering the risk of losing a position while taking parental leave, organizational policies could regulate the hiring of substitutes to replace workers, as well as structured return programs (with phased return schedules, mentorship, training, and reintegration support) for mothers, as proposed by Bowyer et al. [30] and Halrynjo and Mangset [50].

##### Building a Supportive Organizational Culture for Mothers

Organizations need to build a supportive work culture that values work–life balance, diverse identities, personal well-being, and cultivates positive attitudes toward mothers and caregivers of any gender. We suggest that this could be done by inviting mothers and caregivers returning from parental leave to share their experiences, recognizing soft skills gained in parenthood and guardianship and encouraging leaders to openly express support for employees’ family commitments, explicitly manifesting how these responsibilities are known and valued [59,68].

We recommend structured anti-discrimination and anti-harassment policies with anonymous reporting systems, as these are pivotal to foster transparency and accountability, building a safer work environment for women in response to the motherhood penalty identified by Härkönen et al. [60] and Benard and Correll [1]. These policies could also be followed by performance evaluations relative to opportunity, since negative views towards mothers are commonly reflected in lower salaries and reduced advancement opportunities, as shown by Bear and Glick [46] and Kristensen et al. [63]. While implicit bias training may raise awareness of gender stereotypes that affect the perception of mothers, considerate evaluation practices can help reduce inequities in salaries, performance reviews, promotions, and recognition.

We emphasize that performance evaluation practices need to be carefully planned. Ignoring caregiving status by evaluating mothers based only on their performance increases inequalities and preserves disproportional systemic challenges. Considering that mothers are often primarily responsible for domestic labor, face more career interruptions, have reduced work hours or double-shifts, and deviate from traditional career paths, performance assessments should take such factors into account and use specific metrics for mothers based on their context, responsibilities, and challenges while also recognizing the skills and experiences gained through caregiving [50].

These initiatives can be complemented by a more collaborative approach to work. Cross-training programs within teams, collective resources, and collaboration platforms could enable the sharing of experiences and foster a more proportional distribution of demands, responsibilities, and rewards, as proposed by Kibelloh and Bao [69] and Halrynjo and Mangset [50]. This could facilitate transitions and reduce the risk of job loss during parental leave, in addition to dismantling traditional hierarchies and promoting inclusivity. However, this needs to be analyzed in terms of the nature of the job, as in some cases, limited autonomy may be a barrier for mothers’ career advancement, according to Hancioglu and Hartmann’s [70] findings.

Academic careers are especially challenging to new mothers, given the reward system focused on productivity and publication records that place significant constraints on their career progression. We urge a re-evaluation of the criteria for recognition, tenure, and promotion, since academic mothers are commonly unable to comply with the same standards as their male counterparts given their often disproportionate caregiving and domestic responsibilities compared to their male counterparts. Therefore, institutions should promote fairness in the evaluation process and rethink the role of productivity with employees’ caregiving responsibilities in mind. This can be accomplished by accommodating interrupted publication records, extending evaluation periods, or recognizing diverse forms of scholarly outputs other than publications, such as data sharing, teaching, collaborative projects, and community-based interventions [30,36].

More generally, each organization should analyze the specific conditions of its employees, since women with dependent children are not a homogeneous group (e.g., number and age of children are related to different career outcomes [33,70]). This follows Oliveira-Silva and Barbosa-Silva’s [7] recommendations for tailored interventions targeting audience-specific needs and characteristics.

##### The Role of Childcare-Related Government Legislation

The economic costs associated with having children are one of the main difficulties faced by mothers, especially single mothers and those from disadvantaged backgrounds. Therefore, governments need to structure legislation to guarantee mothers’ rights, alleviating the challenges associated with childcare. This could be done by providing affordable, reliable, and accessible childcare services, or by offering tax credits for childcare expenses.

Financial constraints not only increase the likelihood of withdrawal from the labor force but also compel mothers to take jobs of lower status and salaries, which directly hinder their career, as stated by Ma [56] and Hancioglu and Hartmann [70]. Therefore, we emphasize the need for social safety nets and government programs that offer financial support, healthcare, food security, and housing assistance for low-income families, added to reemployment programs. These are especially important during economic crises, as shown by Staniscuaski et al. [44] and Cha [71].

Legislation that guarantees rights for part-time and short-term contract workers is also important, considering that many women from lower social classes are obliged to invest in such career arrangements after motherhood [33,36]. To ensure poorer mothers are not being even more marginalized, we recommend governmental interventions to focus on providing protections against job loss, as well as fair treatment and mechanisms for transitioning contract workers to permanent roles. Another important issue that should be government-level regulated is breastfeeding, with public policies including the creation of lactation spaces [32,72]. Legislation for equal pay is also important to guarantee that mothers are being fairly compensated for their work and to avoid deepening the gender pay gap, as shown by Bear and Glick [46].

Education plays a crucial role in addressing motherhood penalties, so providing specific training programs and learning resources are important strategies to improve mothers’ employability, keep them updated on industry trends, and facilitate their return after career breaks, as proposed by Ma [56] and Härkönen et al. [60]. However, we highlight that education should not target everybody—not only mothers—especially when involving awareness campaigns that challenge stigmas associated with motherhood. This could be achieved by sharing and underscoring successful stories of working mothers, bringing attention to the systemic barriers and stereotypes faced by them, or communicating the benefits associated with family-friendly policies. By educating people about the challenges and contributions of mothers, it is possible to encourage empathy and support from colleagues, employers, and society in general [48,51].

Work in academia should also be targeted by national initiatives that reshape evaluation metrics, recognize outputs such as teaching and service contributions, provide funding for hiring research assistance (which can help mothers with the academic workload), postpone deadlines, and provide flexible work arrangements to mothers, as suggested by Maxwell et al. [59] and Eren [36]. Additionally, public funding agencies need to finance and support research on gender biases or working mothers’ experiences as means of obtaining evidence to inform public policies and legislations.

## 8. Limitations and Recommendations for Future Research

This review is limited by its exclusive analysis of English-language papers, potentially excluding valuable studies in other languages and introducing cultural bias by emphasizing perspectives from English-speaking countries and scholars. Additionally, the 10-year timeframe chosen for analysis, while intending to offer a contemporary perspective, does not allow us to examine historical shifts in workplace dynamics and societal attitudes toward motherhood. Future investigations might consider analyzing longer-term trends, exploring the influence of time, changes in media representations, and the impact of different generations on the career trajectories of mothers. This approach could provide a more nuanced understanding of the complex interplay between societal changes and the career experiences of mothers.

Despite its methodological limitations, this scoping review evidenced important gaps in the scholarship about motherhood and career progression. The lack of intersectionality must be stressed as a relevant characteristic of the reviewed literature. Some studies did not provide basic intersectional information about their participants, such as socioeconomic status, race, ethnicity, class, gender identity, sexuality, or marital status, in the main analysis. Not surprisingly, the reviewed literature presented a predominantly heteronormative perspective about motherhood by overlooking the experience of mothers with diverse sexual orientations or gender identities. Furthermore, the presence of a (typically male) partner was often presumed, leaving the experiences of single mothers under-examined. An additional gap was the lack of studies considering the broader macroeconomic context in which the participant mothers were situated. This is essential, as economic crises are directly related to job availability, salaries, and advancement opportunities, such that financial struggles can also stimulate mothers to return earlier from maternity leave and work for long hours.

Accessing sample characteristics, especially with regard to if and how those from minoritized groups are included, is crucial to design appropriate interventions and unfold theoretical implications. Considering the intersectionalities different women face around the world is mandatory, especially because career progression may change significantly depending on the mother’s background.

An important aspect that future studies should consider is the representation of women among the authors of studies about motherhood. It would be insightful, for example, to investigate how many authors self-identify as women. This could shed light on how much women, more than men, are engaging with and contributing to the discussion and investigation of this topic. Understanding the publishing experiences of researchers who study topics related to motherhood and gender inequality could provide insights into the biases and obstacles that exist within academic publishing systems. Additionally, it could prompt further examination of the barriers and challenges that women face in academic publishing, especially in fields where the motherhood penalty is particularly pronounced.

Therefore, future research on motherhood and career should be more intersectional and inclusive, with a greater emphasis on contextual and structural inequalities when addressing interventions for career progression. Diverse family configurations (e.g., lesbian and gay couples, single mothers, non-monogamous families) and multiple cultural and social factors must be considered as often as possible, as the sexual division of labor can vary depending on cultural values and how mothers are socially perceived.

## 9. Conclusions

This scoping review provided a deeper understanding of how the literature has investigated the impact of motherhood on women’s career progression. Most of the identified impacts were negative and corroborated the motherhood penalty, with concrete barriers to career advancement and leadership. On the other hand, we also identified positive aspects of the relationship between motherhood and career outcomes, usually related to mothers developing more interpersonal, time management, and problem-solving skills.

Despite our attempt to analyze the literature regarding career progression instead of leadership, we were still confronted with an androcentric view. Even the career progress notion was shown to be highly influenced by a typical masculine model of work. In addition to perpetuating a narrow view of what it means to achieve career progression, this fosters gender stereotypes and limits the understanding of diverse career trajectories, especially those pursued by women, working mothers, and other minorities.

Considering how attitudes, feelings, and behaviors resulting from motherhood are mainly caused by situations which women have little control over and those that result from the interaction between work responsibilities and childcare availability, the role of interventions is essential. However, they must focus not only on micro-level actions but also cover the multifaceted dimensions that affect motherhood and care work in workplaces and societies. Therefore, this scoping review provided an updated perspective on a stubborn issue, offering practical insights to encompass its complexities and contribute to a more inclusive and structural understanding of the career trajectories of working mothers.

## Figures and Tables

**Figure 1 behavsci-14-00275-f001:**
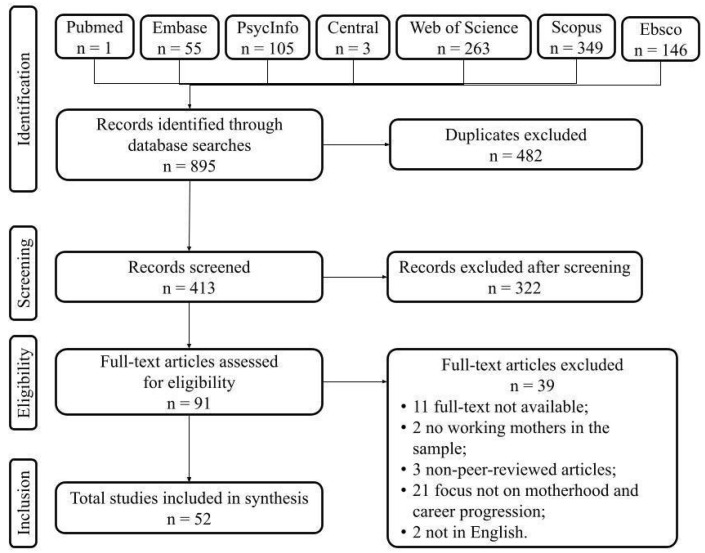
Stages of data collection for the literature review.

**Table 1 behavsci-14-00275-t001:** Characteristics of the included studies.

Categories	n	%
Nationality of the 1st author		
USA	15	28.8%
Germany	7	13.5%
Others (e.g., Sweden, UK, Norway, Austria, Republic of Korea)	30	57.7%
Journal origin		
United Kingdom	23	44.2%
USA	15	28.8%
Others (e.g., Netherlands, Switzerland, Germany)	14	26.9%
Publishing journals		
Gender, Work & Organization	5	9.6%
Acta Sociologica	2	3.8%
Advances in Life Course Research	2	3.8%
Gender & Society	2	3.8%
Gender in Management: An international journal	2	3.8%
Labour Economics	2	3.8%
Social Sciences	2	3.8%
Research design		
Quantitative	25	48.1%
Qualitative	18	34.6%
Mixed methods	9	17.3%
Research design		
Cross-sectional	28	53.8%
Experiment	21	40.4%
Longitudinal	2	3.8%
Data collection method		
Survey	23	44.2%
Interview	33	63.5%
Sample composition		
Female	29	55.8%
Mixed gender	23	44.2%
Context		
Organization	27	51.9%
Academia	18	34.6%
Not reported	7	13.5%
Knowledge field		
No distinction	32	61.5%
STEM	4	7.7%
Medicine	3	5.8%
Others (e.g., tourism and hospitality, accounting, advertising, education)	13	25.0%

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
