# Peer review of "The Impact of Motherhood on Women’s Career Progression: A Scoping Review of Evidence-Based Interventions"

_behavsci, 2024, doi:10.3390/bs14040275_

Round 1

Reviewer 1 Report

Comments and Suggestions for Authors

I am pleased that I had the opportunity review this paper on the motherhood penalty and career progression. One of the key contributions is the disambiguation of career progression from leadership- that subtle change in operational definition keeps the conversation firmly on outcomes in the workplace, rather than in personality traits related to psychological gender and leadership style (which is where I spend most of my time, admittedly). It is because the authors took this perspective that I am in support of it making it to the literature officially and, in turn, become part of the next stages of the conversation related to systemic interventions. And these interventions will only serve to help attenuate the wage gaps that come from existing gender inequalities. 

Speaking of inequalities, I wonder what the authors think about the motherhood penalty as it related to academic publishing? E.g., I wonder how many of the authors in the final 52 studies self-identified as women? Was the road to getting that data into the literature more difficult, such as how many rejections or revisions did it take to get the article into print? How many graduate students or junior faculty were/are counseled to avoid this kind of research because it will not serve one's career progression in academia. I don't think the manuscript has to include responses to these questions in order to be suitable for publication but it could add to the discussion in general. 

Thank you again for exploring this perennially relevant issue and for presenting it so cogently.

Reviewer 2 Report

Comments and Suggestions for Authors

1) Why did you choose to do a scoping review and not a systematic review, also considering the limitations of the scoping review? Perhaps specify.

2) I would suggest reporting the characteristics of the included studies according to the PICOS scheme: participants, interventions, comparisons, outcomes and study design. 

3) It is not clear which studies cited are those that are only part of the review and which are other literature (the PICOS scheme helps). In the introduction and paragraph 2, literature studies supporting the topic and research questions should be cited, without citing the articles included in the review. These should be part of the findings and discussions, to which other literature studies not included in the review should be added. The literature should perhaps be expanded. 

4) I would expand the results part, e.g. by including the total number of participants included in the studies and by writing the results in Table 1 and 2 of the supplementary materials in a more or less discursive way. it would be easier for the reader to have an initial summary. 
